# Why be adversarial? Let's cooperate!:
# Cooperative Dataset Alignment via JSD Upper Bound

**Wonwoong Cho** [* 1]   **Ziyu Gong** [* 1]   **David I. Inouye** [1]

## Abstract

Unsupervised dataset alignment estimates a transformation that maps two or more source domains to a shared aligned domain given only the domain datasets. This task has many applications including generative modeling, unsupervised domain adaptation, and socially aware learning. Most prior works use adversarial learning (i.e., min-max optimization), which can be challenging to optimize and evaluate. A few recent works explore non-adversarial flow-based (i.e., invertible) approaches, but they lack a unified perspective. Therefore, we propose to unify and generalize previous flow-based approaches under a single non-adversarial framework, which we prove is equivalent to minimizing an upper bound on the Jensen-Shannon Divergence (JSD). Importantly, our problem reduces to a min-min, i.e., cooperative, problem and can provide a natural evaluation metric for unsupervised dataset alignment. We present preliminary results of our framework on simulated and real-world data.

## 1. Introduction

In many cases, a practitioner has access to multiple related but distinct datasets such as agricultural measurements from two farms, experimental data collected in different months, or sales data before and after a major event. *Unsupervised* dataset alignment (UDA) is the ML task aimed at aligning these related but distinct datasets in a shared space, which may be a latent space, *without* any pairing information between the two domains (i.e., unsupervised). This task has many applications such as generative modeling (e.g., (Zhu et al., 2017)), unsupervised domain adaptation (e.g., (Grover et al., 2020; Hu et al., 2018)), batch

---
[*]Equal contribution [1]School of Electrical and Computer Engineering, Purdue University. Correspondence to: David I. Inouye <dinouye@purdue.edu>.

Third workshop on *Invertible Neural Networks, Normalizing Flows, and Explicit Likelihood Models* (ICML 2021). Copyright 2021 by the author(s).

effect mitigation in biology (e.g., (Haghverdi et al., 2018)), and fairness-aware learning (e.g., (Zemel et al., 2013)).

The most common approach for obtaining such alignment transformations stems from Generative Adversarial Networks (GAN) (Goodfellow et al., 2014), which can be viewed as minimizing a *lower bound* on the Jensen-Shannon Divergence (JSD) between real and generated distributions. The lower bound is tight if and only if the inner maximization is solved perfectly. CycleGAN (Zhu et al., 2017) maps between *two* datasets via two GAN objectives between the two domains and a cycle consistency loss, which encourages approximate invertibility of the transformations. However, adversarial learning can be quite challenging in practice (see e.g. (Lucic et al., 2018; Kurach et al., 2019)) because of the competitive nature of the min-max optimization problem. Also, the research community only has reasonable model evaluation metrics for certain data types. Specifically, the commonly accepted Frechet Inception Distance (FID) (Heusel et al., 2017) is only applicable to image or auditory data, which have standard powerful pretrained classifiers, and even the implementation of FID can have evaluation issues (Parmar et al., 2021). No clear metrics exist for tabular data or non-perceptual data.

Recently, flow-based methods that leverage invertible models have been proposed for the UDA task (Grover et al., 2020; Usman et al., 2020). AlignFlow (Grover et al., 2020) leverages invertible models to make the model cycle-consistent (i.e., invertible) *by construction* and introduce exact log-likelihood loss terms derived from standard flow-based generative models as a complementary loss terms to the adversarial loss terms. Yet, AlignFlow still leverages adversarial learning and does not provide a general evaluation metric. Log-likelihood ratio minimizing flows (LRMF) (Usman et al., 2020) use invertible flow models and density estimation to avoid adversarial learning altogether and define a new metric based on the log-likelihood ratio. However, LRMF depends heavily on the density model class and can only partially align datasets if the target distribution is not in the chosen density model class. Additionally, the LRMF metric depends on this density model class and is only defined for two datasets.

Therefore, to avoid challenging adversarial learning and

generalize previous flow-based approaches, we propose a unified non-adversarial UDA framework, which we prove is equivalent to minimizing an upper bound on the JSD. Importantly, our problem reduces to a min-min, i.e., *cooperative*, problem, and the JSD upper bound can provide a natural evaluation metric for UDA that can be applied in any domain. Our framework requires two parts, the outer minimization requires an invertible model and the inner minimization requires a density model (e.g., Gaussian mixture models or normalizing flows (Dinh et al., 2017)). We summarize our contributions as follows:

- We prove that a minimization problem over density models is an *upper bound* on a generalized version of JSD that allows for more than two distributions. Importantly, we also theoretically quantify the bound gap and show that it can be made tight if the density model class is flexible enough.

- We use this JSD upper bound to derive a novel regularized loss function for UDA and explain its relationship to prior methods.

- We demonstrate the feasibility of our method on simulated and real-world data.

**Notation**   We will denote distributions as $P_X(\boldsymbol{x})$ where $X$ is the corresponding random variable. Invertible functions will be denoted by $T(\cdot)$. We will use $X_j \sim P_{X_j}$ to denote the observed random variable from the $j$-th distribution. We will use $Z_j \triangleq T_j(X_j) \sim P_{Z_j} \equiv P_{T_j(X_j)}$ to denote the latent random variable of the $j$-th distribution after applying $T_j$ to $X_j$ (and note that $X_j = T_j^{-1}(Z_j)$). We will denote the mixtures of these observed or latent distributions as $P_{X_{\mathrm{mix}}} \triangleq \sum_j w_j P_{X_j}$ and $P_{Z_{\mathrm{mix}}} \triangleq \sum_j w_j P_{Z_j}$, where $\boldsymbol{w}$ is a probability vector. We denote KL divergence, entropy, and cross entropy as $\mathrm{KL}(\cdot, \cdot)$, $\mathrm{H}(\cdot)$, and $\mathrm{H_c}(\cdot, \cdot)$, respectively, where $\mathrm{KL}(P, Q) = \mathrm{H_c}(P, Q) - \mathrm{H}(P)$.

## 2. Regularized Alignment Upper Bound Loss

We first remind the reader of the generalized Jensen-Shannon divergence for more than two distributions, where the standard JSD is recovered if $w_1 = w_2 = 0.5$.

**Definition 1** (Generalized Jensen-Shannon Divergence (GJSD) (Lin, 1991)). *Given $k$ distributions $\{P_{X_j}\}_{j=1}^k$ and a corresponding probability weight vector $\boldsymbol{w}$, the generalized Jensen-Shannon divergence is defined as (proof of equivalence in appendix):*

$$\mathrm{GJSD}_{\boldsymbol{w}}(P_{X_1}, \cdots, P_{X_k}) \triangleq \sum_j w_j \mathrm{KL}(P_{X_j}, \sum_j w_j P_{X_j})$$
$$\equiv \mathrm{H}\left(\sum_j w_j P_{X_j}\right) - \sum_j w_j \mathrm{H}(P_{X_j}). \qquad (1)$$

The goal of distribution alignment is to find a set of transformations $\{T_j(\cdot)\}_{j=1}^k$ (which will be invertible in our

case) such that the latent distributions align, i.e., $P_{T_j(X_j)} = P_{T_{j'}(X_{j'})}$ or equivalently $P_{Z_j} = P_{Z_{j'}}$ for all $j \neq j'$. Given the properties of divergences, this alignment will happen if and only if $\mathrm{GJSD}(P_{Z_1}, \cdots, P_{Z_k}) = 0$. Thus, ideally, we would minimize GJSD directly with respect to $T_j$, i.e.,

$$\min_{T_1, \cdots, T_k \in \mathcal{T}} \mathrm{GJSD}(P_{T_1(X_1)}, \cdots, P_{T_k(X_k)}) \qquad (2)$$
$$\equiv \min_{T_1, \cdots, T_k \in \mathcal{T}} \mathrm{H}\left(\sum_j w_j P_{T_j(X_j)}\right) - \sum_j w_j \mathrm{H}(P_{T_j(X_j)}),$$

where $\mathcal{T}$ is a class of invertible functions.

### 2.1. GJSD Upper Bound

However, we cannot evaluate the entropy terms in Eqn. 2 because we do not know the density of $P_{X_j}$; we only have samples from $P_{X_j}$. Therefore, we will upper bound the first entropy term in Eqn. 2 ($\mathrm{H}\left(\sum_j w_j P_{X_j}\right)$) using an auxiliary density model and decompose the other entropy terms by leveraging the change of variables formula for invertible functions.

**Theorem 1** (GJSD Upper Bound). *Given an auxiliary density model class $\mathcal{Q}$, we form a GJSD upper bound:*

$$\mathrm{GJSD}_{\boldsymbol{w}}(P_{Z_1}, \cdots, P_{Z_k})$$
$$\leq \min_{Q \in \mathcal{Q}} \mathrm{H_c}(P_{Z_{\mathrm{mix}}}, Q) - \sum_j w_j \mathrm{H}(P_{Z_j}),$$

*where the bound gap is exactly $\min_{Q \in \mathcal{Q}} \mathrm{KL}(P_{Z_{\mathrm{mix}}}, Q)$.*

*Proof of Theorem 1.*   For any $Q \in \mathcal{Q}$, we have the following upper bound:

$$\mathrm{GJSD}_{\boldsymbol{w}}(P_{Z_1}, \cdots, P_{Z_k})$$
$$= \underbrace{\mathrm{H_c}(P_{Z_{\mathrm{mix}}}, Q) - \mathrm{H_c}(P_{Z_{\mathrm{mix}}}, Q)}_{=0} + \mathrm{H}(P_{Z_{\mathrm{mix}}}) - \sum_j w_j \mathrm{H}(P_{Z_j})$$
$$= \mathrm{H_c}(P_{Z_{\mathrm{mix}}}, Q) - \mathrm{KL}(P_{Z_{\mathrm{mix}}}, Q) - \sum_j w_j \mathrm{H}(P_{Z_j})$$
$$\leq \mathrm{H_c}(P_{Z_{\mathrm{mix}}}, Q) - \sum_j w_j \mathrm{H}(P_{Z_j}),$$

where the inequality is by the fact that KL divergence is non-negative and the bound gap is equal to $\mathrm{KL}(P_{Z_{\mathrm{mix}}}, Q)$. The $Q$ that achieves the minimum in the upper bound is equivalent to the $Q$ that minimizes the bound gap, i.e.,

$$Q^* = \arg\min_{Q \in \mathcal{Q}} \mathrm{H_c}(P_{Z_{\mathrm{mix}}}, Q) \underbrace{- \sum_j w_j \mathrm{H}(P_{Z_j})}_{\text{Constant w.r.t. } Q} \qquad (3)$$

$$= \arg\min_{Q \in \mathcal{Q}} \mathrm{H_c}(P_{Z_{\mathrm{mix}}}, Q) \underbrace{- \mathrm{H}(P_{Z_{\mathrm{mix}}})}_{\text{Constant w.r.t. } Q} \qquad (4)$$

$$= \arg\min_{Q \in \mathcal{Q}} \mathrm{KL}(P_{Z_{\mathrm{mix}}}, Q). \qquad (5)$$

$\square$

The tightness of the bound depends on how well the class of density models $\mathcal{Q}$ (e.g., mixture models, normalizing

flows, or autoregressive densities) can approximate $P_{Z_{\text{mix}}}$; notably, the bound can be made tight if $P_{Z_{\text{mix}}} \in \mathcal{Q}$. Also, one key feature of this upper bound is that the cross entropy term can be evaluated using only samples from $P_{X_j}$ and the transformations $T_j$, i.e., $\text{H}_\text{c}(P_{Z_{\text{mix}}}, Q) = \sum_j w_j \mathbb{E}_{P_{X_j}}[-\log Q(T_j(\boldsymbol{x}_j))]$. However, we still cannot evaluate the other entropy terms $\text{H}(P_{Z_j})$ since we do not know the densities of $P_{Z_j}$ (or $P_{X_j}$). Thus, we leverage the fact that the $T_j$ functions are invertible to define an entropy change of variables.

**Lemma 2** (Entropy Change of Variables). *Let $X \sim P_X$ and $Z \triangleq T(X) \sim P_Z$, where $T$ is an invertible transformation. The entropy of $Z$ can be decomposed as follows:*

$$\text{H}(P_Z) = \text{H}(P_X) + \mathbb{E}_{P_X}\left[\log |J_T(\boldsymbol{x})|\right], \quad (6)$$

*where $|J_T(\boldsymbol{x})|$ is the determinant of the Jacobian of $T$.*

The key insight from this lemma is that $\text{H}(P_X)$ is a constant with respect to $T$ and can thus be ignored when optimizing $T$, while $\mathbb{E}_{P_X}[\log |J_T(\boldsymbol{x})|]$ can be approximated using only samples from $P_X$. Combining Theorem 1 and Lemma 2, we can arrive at our final objective function which is equivalent to minimizing an upper bound on the GJSD:

$$\begin{aligned}
\text{GJSD}_{\boldsymbol{w}}&(P_{Z_1}, \cdots, P_{Z_k}) \\
&\leq \min_{Q \in \mathcal{Q}} \text{H}_\text{c}(P_{Z_{\text{mix}}}, Q) - \sum_j w_j \text{H}(P_{Z_j}) \quad (7) \\
&= \min_{Q \in \mathcal{Q}} \sum_j w_j \mathbb{E}_{P_{X_j}}[-\log Q(T_j(\boldsymbol{x}))|J_{T_j}(\boldsymbol{x})|] \\
&\qquad - \sum_j w_j \text{H}(P_{X_j}),
\end{aligned} \quad (8)$$

where the last term $-\sum_j w_j \text{H}(P_{X_j})$ is constant with respect to $T_j$ functions so they can be ignored. We formally define this loss function as follows.

**Definition 2** (Alignment Upper Bound Loss). *Given $k$ continuous distributions $\{P_{X_j}\}_{j=1}^k$, a class of continuous distributions $\mathcal{Q}$, and a probability weight vector $\boldsymbol{w}$, the alignment upper bound loss is defined as follows:*

$$\begin{aligned}
\mathcal{L}_{\text{AUB}}&(T_1, \cdots, T_k; \{P_{X_j}\}_{j=1}^k, \mathcal{Q}, \boldsymbol{w}) \\
&\triangleq \min_{Q \in \mathcal{Q}} \sum_j w_j \mathbb{E}_{P_{X_j}}[-\log |J_{T_j}(\boldsymbol{x})| Q(T_j(\boldsymbol{x}))], \quad (9)
\end{aligned}$$

*where $T_j$ are invertible and $|J_{T_j}(\boldsymbol{x})|$ is the absolute value of the Jacobian determinant.*

Notice that this alignment loss can be seen as learning the best base distribution given fixed flow models $T_j$. We now consider the theoretical optimum if we optimize over all invertible functions.

**Theorem 3** (Alignment at Global Minimum of $\mathcal{L}_{\text{AUB}}$). *If $\mathcal{L}_{\text{AUB}}$ is minimized over the class of all invertible functions, a global minimum of $\mathcal{L}_{\text{AUB}}$ implies that the latent distributions are aligned, i.e., $P_{T_j(X_j)} = P_{T_{j'}(X_{j'})}$ for all $j \neq j'$. Notably, this result holds regardless of $\mathcal{Q}$.*

Informally, this can be proved by showing that the problem decouples into separate normalizing flow losses where $Q$ is the base distribution and the optimum is achieved only if $P_{T_j(X_j)} = Q$ for all $T_j$ (formal proof in the appendix). This alignment of the latent distributions also implies the translation between any of the *observed* component distributions. The proof follows directly from Theorem 3 and the change of variables formula.

**Corollary 4** (Translation at Global Minimum of $\mathcal{L}_{\text{AUB}}$). *Similar to Theorem 3, a global minimum of $\mathcal{L}_{\text{AUB}}$ implies translation between any component distributions using the inverses of $T_j$, i.e., $P_{T_{j'}^{-1}(T_j(X_j))} = P_{X_{j'}}$ for all $j \neq j'$.*

### 2.2. Regularization via Transportation Cost

While the alignment objective is the most challenging part of UDA, we argue that regularization is also critical for practical and stable alignment (or translation) between datasets because there are many optimal alignment solutions—even infinitely many in most cases (see appendix for two examples). We alleviate this issue by adding expected transportation cost (usually squared Euclidean distance) as a regularization to our objective inspired by optimal transport (OT) concepts.

**Definition 3** (Regularized Alignment Upper Bound Loss). *Given similar setup as in Def. 2 and a transportation cost function $c(a, b) \geq 0$ for transporting a point from $a$ to $b$, the regularized alignment upper bound loss is defined as:*

$$\begin{aligned}
\mathcal{L}_{\text{RAUB}}&(T_1, \cdots, T_k; \{P_{X_j}\}_{j=1}^k, \mathcal{Q}, \boldsymbol{w}, \lambda, c) \\
&\triangleq \min_{Q \in \mathcal{Q}} \sum_j w_j \mathbb{E}_{P_{X_j}}[-\log |J_{T_j}(\boldsymbol{x})| Q(T_j(\boldsymbol{x})) \quad (10) \\
&\qquad\qquad + \lambda c(\boldsymbol{x}, T_j(\boldsymbol{x}))].
\end{aligned}$$

### 2.3. Relationship to Prior Works

**AlignFlow is special case without adversarial terms** AlignFlow (Grover et al., 2020) *without* adversarial loss terms is a special case of our method for two distributions where the density model class $\mathcal{Q}$ only contains the standard normal distribution (i.e., a singleton class) and no regularization is used (i.e., $\lambda = 0$). Thus, AlignFlow can be viewed as initially optimizing a poor upper bound on JSD; however, the JSD bound becomes tighter as training progresses because the latent distributions *independently* move towards the same normal distribution.

**LRMF is special case with only one transformation** Log-likelihood ratio minimizing flows (LRMF) (Usman et al., 2020) is also a special case of our method for only two distributions, where one transformation is fixed at the identity (i.e., $T_2 = \text{Id}$) and no regularization is applied (i.e., $\lambda = 0$). While the final practical LRMF objective is a special case of ours, the theory is developed from a different

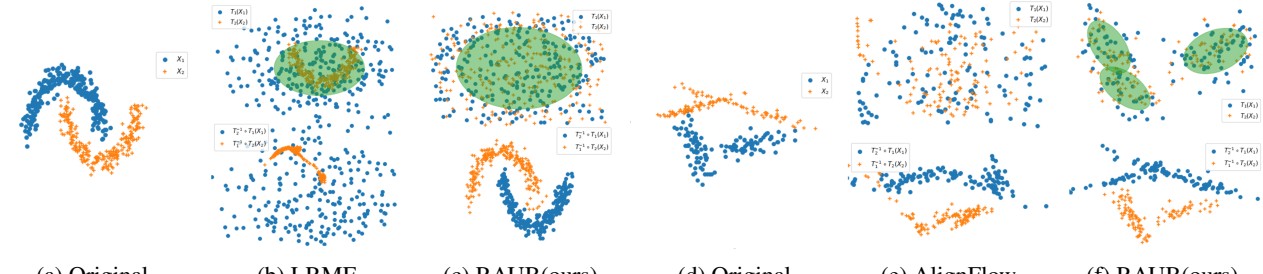

(a) Original     (b) LRMF     (c) RAUB(ours)     (d) Original     (e) AlignFlow     (f) RAUB(ours)

Figure 1: Top row is latent space and bottom is the data translated into the other space. (a-c) LRMF, which only has *one* transformation $T$ may not be able to align the datasets if the density model class $\mathcal{Q}$ is not expressive enough (in this case Gaussian distributions) while using two transformations as in our framework can align them. (d-f) AlignFlow (without adversarial terms) may not align because $Q_z$ is fixed at a standard normal, while our approach with learnable mixture of Gaussians for $Q_z$ is able to learn an alignment (both use the same $T_j$ models).

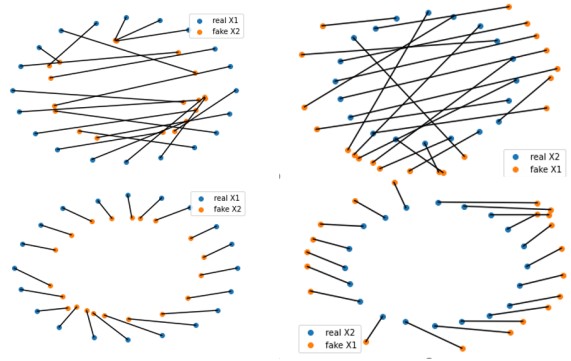

(a) Paired transform for $X_1$     (b) Paired transform for $X_2$

Figure 2: An unregularized alignment loss (top) can lead to excessive and unexpected movement of points in the latent representation (lines connect transported points), while our regularized alignment loss (bottom) yields a unique and regularized solution that moves points significantly less and is closer to the identity function.

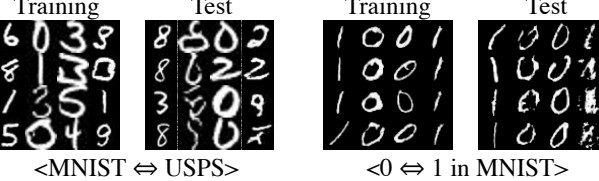

Training    Test      Training    Test

<MNIST ⇔ USPS>      <0 ⇔ 1 in MNIST>

Figure 3: Preliminary results on a high dimensional real world datasets demonstrate the feasibility of our method. The complex translation between MNIST and USPS datasets (left) does not seem to overfit, while the simple translation between MNIST 0 to 1 may overfit (as seen by right most column of test). The first and third columns are the original images, and the second and the fourth columns are the translated images.

but complementary perspective. The LRMF metric developed requires an assumption about a given density model class, which enables a zero point (or absolute value) of the metric to be estimated but requires fitting extra domain density models. Usman et al. (2020) also do not uncover the connection of the objective as an upper bound on JSD re-

gardless of the density model class. Additionally, to ensure alignment, LRMF requires that the density model class includes the true target distribution because only one invertible transform is used, while our approach can theoretically align even if the shared density model class is weak (see Theorem 3 and our simulated experiments).

**Cooperative versus Adversarial Networks** Analogous to the generator $G$ and the discriminator $D$ in adversarial learning, our framework has two main networks, $T_j$ and $Q_z$. We can use *any* invertible function for $T_j$ (e.g., coupling-based flows (Dinh et al., 2017), neural ODE flows (Grathwohl et al., 2018), or residual flows (Chen et al., 2019)) and *any* (approximate) density models for $Q_z$ (e.g., kernel densities (in low dimensions), mixture models, autoregressive densities (Salimans et al., 2017), normalizing flows (Kingma and Dhariwal, 2018), or even VAEs (Kingma and Welling, 2019)). Thus, our framework has similar modularity compared to adversarial approaches. In contrast, we have a min-min, i.e., cooperative, optimization problem, but our transformations must be invertible and the auxiliary density model $Q_z$ may be more difficult to train than the auxiliary discriminator $D$. We expect these limitations to be alleviated as new invertible models and density models are continually being developed.

## 3. Experiments and Conclusion

We first demonstrate the differences of our approach to LRMF and AlignFlow in Fig. 1 and the importance of regularization in Fig. 2. We also demonstrate the feasibility of our approach for high-dimensional datasets in some preliminary experiments shown in Fig. 3. Please see appendix for details for experiments. Yet, scaling up our framework in practice is still a fundamental challenge, and our approach inherits some weaknesses of JSD, which may not give useful gradient information in certain cases. Thus, while these experiments and our theoretical work build a unified foundation for cooperative alignment learning, our work also open up new theoretical and practical questions.

## Acknowledgement

The authors acknowledge support from the Army Research Lab through Contract number W911NF-2020-221.

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

# A. Proofs

*Proof of Equivalence in Def. 1.* While the proof of the equivalence is well-known, we reproduce here for completeness. As a reminder, the KL divergence is defined as:

$$\mathrm{KL}(P,Q) = \mathbb{E}_P[\log \tfrac{P(x)}{Q(x)}] = \mathbb{E}_P[-\log Q(x)] - \mathbb{E}_P[-\log P(x)] = \mathrm{H_c}(P,Q) - \mathrm{H}(P), \tag{11}$$

where $\mathrm{H_c}(\cdot,\cdot)$ denotes the cross entropy and $\mathrm{H}(\cdot)$ denotes entropy. Given this, we can now easily derive the equivalence:

$$\mathrm{GJSD}_{\boldsymbol{w}}(P_{X_1}, \cdots, P_{X_k}) = \sum_j w_j \, \mathrm{KL}(P_{X_j}, P_{X_{\mathrm{mix}}}) \tag{12}$$

$$= \sum_j w_j (\mathrm{H_c}(P_{X_j}, P_{X_{\mathrm{mix}}}) - \mathrm{H}(P_{X_j})) \tag{13}$$

$$= \sum_j w_j \, \mathrm{H_c}(P_{X_j}, P_{X_{\mathrm{mix}}}) - \sum_j w_j \, \mathrm{H}(P_{X_j}) \tag{14}$$

$$= \sum_j w_j \mathbb{E}_{P_{X_j}}[-\log P_{X_{\mathrm{mix}}}] - \sum_j w_j \, \mathrm{H}(P_{X_j}) \tag{15}$$

$$= \sum_j w_j \int_{\mathcal{X}} -P_{X_j}(x) \log P_{X_{\mathrm{mix}}}(x) dx - \sum_j w_j \, \mathrm{H}(P_{X_j}) \tag{16}$$

$$= \int_{\mathcal{X}} -\sum_j w_j P_{X_j}(x) \log P_{X_{\mathrm{mix}}}(x) dx - \sum_j w_j \, \mathrm{H}(P_{X_j}) \tag{17}$$

$$= \int_{\mathcal{X}} -P_{X_{\mathrm{mix}}}(x) \log P_{X_{\mathrm{mix}}}(x) dx - \sum_j w_j \, \mathrm{H}(P_{X_j}) \tag{18}$$

$$= \mathrm{H}(P_{X_{\mathrm{mix}}}) - \sum_j w_j \, \mathrm{H}(P_{X_j}). \tag{19}$$

$\square$

*Proof of Lemma 2.* First, we note the following fact from the standard change of variables formula:

$$P_X(\boldsymbol{x}) = P_Z(T(\boldsymbol{x}))|J_T(\boldsymbol{x})|$$
$$\Rightarrow P_X(\boldsymbol{x})|J_T(\boldsymbol{x})|^{-1} = P_Z(T(\boldsymbol{x})). \tag{20}$$

We can now derive our result using the change of variables for expectations (i.e., LOTUS) and the probability change of variables from above:

$$\mathrm{H}(P_Z) = \mathbb{E}_{P_Z}[-\log P_Z(\boldsymbol{z})] = \mathbb{E}_{P_X}[-\log P_Z(T(\boldsymbol{x}))]$$
$$= \mathbb{E}_{P_X}[-\log P_X(\boldsymbol{x})|J_T(\boldsymbol{x})|^{-1}]$$
$$= \mathbb{E}_{P_X}[-\log P_X(\boldsymbol{x})] + \mathbb{E}_{P_X}[-\log |J_T(\boldsymbol{x})|^{-1}]$$
$$= \mathrm{H}(P_X) + \mathbb{E}_{P_X}[\log |J_T(\boldsymbol{x})|].$$

$\square$

*Proof of Theorem 3.* Given any fixed $Q$, minimizing $\mathcal{L}_{\mathrm{AUB}}$ decouples into minimizing separate normalizing flow losses where $Q$ is the base distribution. For each normalizing flow, there exists an invertible $T_j$ such that $T_j(X_j) \sim Q$, and this achieves the minimum value of $\mathcal{L}_{\mathrm{AUB}}$. More formally,

$$\min_{T_1,\cdots,T_k} \mathcal{L}_{\mathrm{AUB}}(T_1, \cdots, T_k) \tag{21}$$

$$= \min_{T_1,\cdots,T_k} \sum_j w_j \mathbb{E}_{P_{X_j}}[-\log |J_{T_j}(\boldsymbol{x})| \, Q(T_j(\boldsymbol{x}))] \tag{22}$$

$$= \sum_j w_j \min_{T_j} \mathbb{E}_{P_{X_j}}[-\log |J_{T_j}(\boldsymbol{x})| \, Q(T_j(\boldsymbol{x}))] + \mathrm{H}(P_{X_j}) - \mathrm{H}(P_{X_j}) \tag{23}$$

$$= \sum_j w_j \min_{T_j} \mathbb{E}_{P_{X_j}}[-\log |J_{T_j}(\boldsymbol{x})| \, Q(T_j(\boldsymbol{x}))] + \mathrm{H}(P_{X_j}) - \mathbb{E}_{P_{X_j}}[-\log P_{X_j}(\boldsymbol{x})]) \tag{24}$$

$$= \sum_j w_j \, \mathrm{H}(P_{X_j}) + \sum_j w_j \min_{T_j} \mathbb{E}_{P_{X_j}}[\log \tfrac{P_{X_j}(\boldsymbol{x})|J_{T_j}(\boldsymbol{x})|^{-1}}{Q(T_j(\boldsymbol{x}))}] \tag{25}$$

$$= \sum_j w_j \, \mathrm{H}(P_{X_j}) + \sum_j w_j \min_{T_j} \mathbb{E}_{P_{X_j}}[\log \tfrac{P_{T_j(X_j)}(T_j(\boldsymbol{x}))}{Q(T_j(\boldsymbol{x}))}] \tag{26}$$

$$= \sum_j w_j \, \mathrm{H}(P_{X_j}) + \sum_j w_j \min_{T_j} \mathbb{E}_{P_{T_j(X_j)}}[\log \tfrac{P_{T_j(X_j)}(\boldsymbol{z})}{Q(\boldsymbol{z})}] \tag{27}$$

$$= \sum_j w_j \, \mathrm{H}(P_{X_j}) + \sum_j w_j \min_{T_j} \mathrm{KL}(P_{T_j(X_j)}, Q). \tag{28}$$

Given that $\mathrm{KL}(P, Q) \geq 0$ and equal to 0 if and only if $P = Q$, the global minimum is achieved only if $P_{T_j(X_j)} = Q, \forall j$ and there exist such invertible functions (e.g., the optimal Monge map between $P_{X_j}$ and $Q$ for squared Euclidean transportation cost (Peyré and Cuturi, 2019)). Additionally, the optimal value is $\sum_j w_j \mathrm{H}(P_{X_j})$, which is constant with respect to the $T_j$ transformations. $\qquad\square$

## B. Examples of Non-Unique Alignment Solutions

### B.1. Gaussian Example

Suppose the component distributions are normal distributions, i.e., $X_1 \sim \mathcal{N}(\mu_1, I)$ and $X_2 \sim \mathcal{N}(\mu_2, I)$, and for even greater simplicity, we assume $T_2$ is the identity, i.e., $T_2(\boldsymbol{x}) = \boldsymbol{x}$. Then, a global optimal solution could be $T_1(\boldsymbol{x}) = U(\boldsymbol{x} - \mu_1 + \mu_2)$ for *any* orthogonal matrix $U$, i.e., there are infinitely many invertible functions that align the distributions. Note that this lack of unique solutions is not restricted to orthogonal rotations (see appendix for a more complex example).

### B.2. Complex Example

Consider the 1D case where $\mathcal{Q}$ only contains the uniform distribution. Thus, $T_1$ and $T_2$ must map their distributions to the uniform distribution for alignment. One solution would be that $T_1 = F_1$ and $T_2 = F_2$ where $F_1$ and $F_2$ are the CDFs of $P_{X_1}$ and $P_{X_2}$. Yet, there are infinitely many other possible solutions. Consider an invertible function that subdivides the unit interval into an arbitrarily large number of equal length intervals and then shuffles these intervals with a fixed arbitrary permutation. More formally, we could define this as:

$$
S_{m,\pi}(x) = \begin{cases} x - \frac{1}{m} + \frac{\pi(1)}{m} & \text{if } x \in [0, \frac{1}{m}) \\ x - \frac{2}{m} + \frac{\pi(2)}{m} & \text{if } x \in [\frac{1}{m}, \frac{2}{m}) \\ \vdots & \vdots \\ x - \frac{m}{m} + \frac{\pi(m)}{m} & \text{if } x \in [\frac{m-1}{m}, 1] \end{cases} ,
\tag{29}
$$

where $\pi(\cdot)$ is a permutation of the integers 1 to $m$. Given this, then other optimal solutions could be $T_1 = S_{m,\pi} \circ F_1$ and $T_2 = F_2$ for any $m > 1$ and any permutation $\pi$. This idea could be generalized to higher dimensions as well by mapping to the multivariate uniform distribution and subdividing the unit hypercube similarly.

## C. 2D dataset comparison with related works

### C.1. Single $T$ vs. Double $T$'s (LRMF vs. Ours)

We first compare our method with LRMF (Usman et al., 2020) method. We construct the experiment to have the task: Transform between the two half-circled distributions $X_1$ and $X_2$ in the moons dataset. In this example, we made two models, one with LRMF setup and one with our RAUB setup. As illustrated in Figure 1, the LRMF method fails to transform between $X_1$ and $X_2$. Even though $Q$ can model well enough for $T_1(X_1)$, $Q$ can only model the mean and variance of $T_2(X_2)$ which is obviously not informative enough. Therefore, the LRMF fails to transform between two datasets. While in the RAUB setup, both $T_1(X_1)$ and $T_2(X_2)$ are modeled to the same distribution while $Q$ can model well enough. And the resulting inverted version of $X_1$ and $X_2$ shows valid transformation. Therefore, the performance of the LRMF model is limited by the power of the density model $Q$ which means if $Q$ fails to model one of the transformed data distribution well enough, data alignment cannot be achieved with high performances.

### C.2. Simple Fixed $Q$ vs. Learnable $Q$ (AlignFlow vs. Ours)

Next we compare our method with AlignFlow(Grover et al., 2020; Hu et al., 2018) setup. We construct the experiment to have the task: Transform between the two random patterns $X_1$ and $X_2$ in the randomly generated datasets. Again, we made two models with AlignFlow and our RAUB setups respectively. As illustrated in Figure 1, the AlignFlow method fails to transform between $X_1$ and $X_2$, because the transformed dataset $T_1(X_1)$ and $T_2(X_2)$ failed to reach the normal distribution $Q$. While in the RAUB setup, the density model $Q$ is learned to help fit the transformed distributions $T_1(X_1)$ and $T_2(X_2)$, which allows them to be aligned with each other. Therefore, the performance of the AlignFlow model is limited by the performances of the invertible functions.

### C.3. Regularized vs. Un-regularized (Some prior works vs. Ours)

We finally show the importance of the regularization term. We construct the experiment to have the task: Transform between two concentric circles with the same mean but slighted different radius. In this example, we make two models with RAUB setup, but one with regularization and one without. As illustrated in the Figure 2, both models are able to transform between two distributions with high performances. However, the transformation pattern is not natural in terms of the 'moving cost' of each point. Each pair created by the unregularized model has bigger transportation cost compared to the pairs created by the regularized model. Therefore, we argue that by adding a transportation cost, the resulting transformation between samples would be closer to an identity transformation and therefore more stable.

**Note:** All implementation details on the toy dataset are available in the Appendix C.

## D. Real-world datasets

To verify the extensibility of our fundamental idea, we also performed our experiments in real-world data. Concretely, we compare our model's performance with the baselines in the image translation task since an evaluation on the dataset alignment are intuitive and interpretable.

### D.1. Experiment Details

We use MNIST and USPS dataset in our experiments. Both two datasets are composed of hand-written digits with 10 classes. Specifically, for the simplest setting, we do our experiments with zero and one classes of MNIST datasest because a translation from zero to one and vice versa requires a semantic transformation, i.e., those two numbers cannot be transformed into each other with a simple translation or rotation. We next perform our experiments with more complicated settings by aligning two different MNIST and USPS datasets.

To cover the high complexity of the real-world data, we exploit a CNN-based flow model (Dinh et al., 2016) as our invertible functions. We also introduce the state of the art density model (Salimans et al., 2017) as our $Q$. Regarding a training procedure, as mentioned in subsection F.1, we first pre-train our $Q_z$ model to efficiently train our $T$ functions. We then set our frameworks to gradually transfer knowledge from $Q$ to $Ts$ by introducing $\beta$ to our loss function, as elaborated in subsection F.2. Learning rate is empirically set to be 0.002 and scheduled to be exponentially decreased after training 10 epoch with a decaying factor of 0.95.

### D.2. Experiment Results

Fig. 3 shows the results of the two real world datasets. The left experiment in the figure shows that the upper bound on the JSD is effectively minimized since the translations between MNIST and USPS are decently working. This implies that common structures of the different domains, e.g., digit information are properly mapped into the shared representations $T_1(X_1)$ and $T_2(X_2)$ while distinctive characteristics of the domains, e.g., a bigger and a thicker pattern of USPS can be transformed via $T_1^{-1}$ and $T_2^{-1}$. We believe this demonstrates that the tight upper bound (by $Q$) of the JSD effectively successively forms the indistinguishable representations.

On the other hand, our proposed idea has some limitations to tackle in the realworld dataset. First of all, as shown in the simpler $0 \Leftrightarrow 1$ experiment in Fig. 3, our complex $T$ with the simple $Q$ yields the overfitting. Second, $\beta$ in the realworld experiments has to be carefully set in training our frameworks. To elaborate, once the gradually increasing $\beta$ reaches a specific value, following the fractional distributions strategy in subsection F.2, it starts to make the model performance worse. Third, model performance with the higher resolution, e.g., $256 \times 256$ needs to be conducted to convincingly verify the performance in realworld dataset. We think these limitations have to be explored to better understand the behavior of our frameworks in realworld dataset.

## E. Detailed Parameters Used in 2D Dataset Experiment

### E.1. LRMF vs. Ours Experiment

- $T$ for LRMF setup: $T_1$: 8 channel-wise mask for Real-NVP model with $s$ and $t$ derived from 64 hidden channels of fully connected networks. $T_2$: Identity function.

- $T$ for RAUB setup: $T_1$ and $T_1$: 8 channel-wise mask for Real-NVP model with $s$ and $t$ derived from 64 hidden channels of fully connected networks. Regularization coefficient $\lambda = 0$

- $Q$ for both: A single Gaussian distribution with trainable mean and trainable variances.

### E.2. Alignflow vs. Ours Experiment

- $T$ for both: 2 channel-wise mask for RealNVP model with $s$ and $t$ derived from 8 hidden channels of fully connected networks.

- $Q$ for Alignflow setup: A single fixed normal distribution.

- $Q$ for RAUB setup: A learnable mixture of Gaussian with 3 components. Regularization coefficient $\lambda = 0$

### E.3. Regularized vs. Unregularized Experiment

1. $T$ for both: 8 channel-wise mask for RealNVP model with $s$ and $t$ derived from 64 hidden channels of fully connected networks.

2. $Q$ for both: A learnable mixture of Gaussian with 2 components.

3. $\lambda$ for unregularized Experiment : $\lambda = 0$

4. $\lambda$ for regularized Experiment : $\lambda = 0$

## F. Algorithm

### F.1. Pre-trained Model

One of the benefits of our method over the baseline methods is that it is possible to harness the power of the pre-trained density model as our $Q$. It is worth mentioning the use of the pre-trained $Q$ enables the gap between the upperbound and the GJSD to be small, thus the better performance can be theoretically ensured compared to baselines with relatively simple $Q$. Furthermore, it is possible to accomplish the statistical and computational efficiency in a training procedure. Specifically, the number of required data to train the networks $T$ can be significantly reduced and the training time can be shortened. Based on these, we leveraged one of the pre-trained state-of-the-art density model () as our $Q$ throughout our experiment.

### F.2. Train with fractional distributions

Most state-of-the-art density would not have a nice convex structure across the entire domain of the images. They will mostly have a nice peak structure within a small range from the true distributions but remain noisy for the rest domain. This would cost the result to fall into local minimum easily at the very beginning and slow or even prevent the loss to a more desirable low value. At the same time, this narrowed range of convexity pattern will make the transform function $T$ much harder to learn so that the transformed distribution $T(x)$ will fail to fit the distribution of $Q_z$. This is also an even bigger problem if we are using a pre-trained model of $Q_z$ to begin with. Therefore, in order to circumvent this kind of situation, we propose to use a fractional power of distributions of $Q_z$ at the first epoch, and slowly increase the power up to 1 to met the original loss objectives.

The basic idea behind the newly introduced fractional power is to expand the variance of $Q_z$ so that the originally nice range of convexity will get expand, so that the algorithm should have a relatively loss function structure to start with. As the training goes, since $T(x)$ is more close to the distribution of density model, we can slightly increase the fractional power to reduce the range of complexity. After we have increased our power to 1, the objective is exactly the same of our original loss function. By using this kind of warm start epochs, we can have our $T(x)$ to start at a relatively good range in $Q_z$, which result in a more efficient learning curve for $T$.

The implementation of this idea is also straightforward: By introducing the fractional power $\beta$, we have $\log |J_{T_j}(x)| Q(T_j)^{\beta}(x) = \log |J_{T_j}(x)| + \log Q(T_j)^{\beta}(x) = \log |J_{T_j}(x)| + \beta \log Q(T_j)(x)$.