# OpenReview forum: "Why be adversarial? Let's cooperate!: Cooperative Dataset Alignment via JSD Upper Bound"
_ICML.cc/2021/Workshop/INNF — INNF+ 2021 spotlighttalk_

### Official Review · Reviewer_FdAo · 2021-06-07

**Rating:** Accept
**Confidence:** 4

**Summary:**

The authors provide a framework for flow-based unsupervised dataset/distribution alignment. They develop a variational objective that is shown to upper bound the Jensen-Shannon Divergence. They show that previous non-adversarial flow-based models can be cast as a special case of their framework. Finally, they provide some promising preliminary results of their method.

**Justification For Rating:**

The idea is clear and the paper is well-written. Seems like a promising research direction.

---

### Official Review · Reviewer_4pES · 2021-06-11

**Rating:** Accept
**Confidence:** 3

**Summary:**

In “Why be adversarial? Let’s cooperate! Cooperative dataset alignment via JSD upper bound,” the authors present a novel technique for unsupervised data alignment. In contrast to adversarial technniques for UDA, normalizing flows are employed for both transformations, enabling a learning framework that minimizes a JSD upper bound, which is shown to be loose using other techniques.  The authors derive valuable results linking the variety of deep techniques for UDA, which motivates their approach.  Visual comparisons to other UDA approaches and preliminary results on MNIST-USPS are given.

**Justification For Rating:**

This paper is an excellent example of how normalizing flows continue to enhance methodology in various domains of machine learning.  The submission is well-written, and contains valuable theorem proofs for UDA that leverage the availability and tractability of normalizing flows.  I highly recommend that this paper be accepted.

The caption of Figure 1 must be edited.

---

### Official Review · Reviewer_fzPG · 2021-06-13

**Rating:** Accept
**Confidence:** 4

**Summary:**

This paper proposes to perform dataset alignment using invertible networks, mapping all data invertibly to a shared latent representation upon which the density of the mapped samples is learned by a shared mixture density model. This is done by rewriting the objective as an upper bound on the Generalized Jensen-Shannon Divergence, and then minimizing this upper bound. This result subsumes and unifies prior likelihood-based approaches to the problem, and provides a more straightforward optimization than adversarial approaches, although it is currently somewhat limited by the representational power of the latent mixture density model.

**Justification For Rating:**

I found this paper to be very well-motivated, including from a theoretical perspective, and believe that the novel ideas presented here will be of strong interest to workshop attendees.

I would like to ask a couple questions of the authors.

1. Why is the Q model more difficult to train than the discriminator D from adversarial approaches? It seems that one of the main motivations here is to _avoid_ difficult training regimes, yet we end up with a Q model that is difficult to optimize.
2. Since the optimal Q model is a mixture distribution, did you try using a discrete mixture of flows (e.g. https://arxiv.org/abs/1907.10448) to parametrize this distribution? Or perhaps a continuous mixture (e.g. https://arxiv.org/abs/1909.13833)? Perhaps these might improve the performance of the model.

---

### Decision · Program_Chairs · 2021-06-14

Accept (spotlight talk)